# Repurposing CRISPR-Cas13 systems for robust mRNA trans-splicing

David N. Fiflis [1], Nicolas A. Rey [2], Harshitha Venugopal-Lavanya [1], Beatrice Sewell[2], Aaron Mitchell-Dick[2], Katie N. Clements[2], Sydney Milo[1], Abigail R. Benkert[2], Alan Rosales[1], Sophia Fergione[2] & Aravind Asokan [1,2,3] ✉

Type VI CRISPR enzymes have been developed as programmable RNA-guided Cas proteins for eukaryotic RNA editing. Notably, Cas13 has been utilized for site-targeted single base edits, demethylation, RNA cleavage or knockdown and alternative splicing. However, the ability to edit large stretches of mRNA transcripts remains a significant challenge. Here, we demonstrate that CRISPR-Cas13 systems can be repurposed to assist trans-splicing of exogenous RNA fragments into an endogenous pre-mRNA transcript, a method termed CRISPR Assisted mRNA Fragment Trans-splicing (CRAFT). Using split reporter-based assays, we evaluate orthogonal Cas13 systems, optimize guide RNA length and screen for optimal trans-splicing site(s) across a range of intronic targets. We achieve markedly improved editing of large 5' and 3' segments in different endogenous mRNAs across various mammalian cell types compared to other spliceosome-mediated trans-splicing methods. CRAFT can serve as a versatile platform for attachment of protein tags, studying the impact of multiple mutations/single nucleotide polymorphisms, modification of untranslated regions (UTRs) or replacing large segments of mRNA transcripts.

The cellular RNA processing machinery has many attributes that can be exploited to manipulate the transcriptome. Specifically, RNA splicing is well conserved in higher eukaryotes and executed by a large ribonucleoprotein complex called the spliceosome. The canonical function of the spliceosome is to catalyze a dual trans-esterification reaction, that joins adjacent exons on the same transcript and removes the intervening intronic sequence; a process referred to as cis-splicing[1]. This splicing machinery has been previously exploited to achieve mRNA trans-splicing, which involves targeted incorporation of recombinant exon(s) into a pre-mRNA transcript. This approach termed spliceosome-mediated RNA trans-splicing or SMaRT features an RNA molecule comprised of an antisense sequence linked to a hemi-intron and one or more exons[2]. Following delivery to the nucleus, the hybridization of the antisense binding domain to the target pre-mRNA enables the intronic sequence to incorporate the recombinant exons in trans by co-opting the splicing machinery, resulting in a chimeric mRNA product. Despite the ability of SMaRT to rewrite multiple kilobases of

target mRNAs, the widespread adoption of this approach as a tool to study RNA biology or therapeutic applications thereof, has generally been hindered by low efficiency[3–6].

Concurrent with the advent of multiple RNA editing approaches, the type VI family of CRISPR systems, which is comprised of single effector RNA-guided, RNA-targeting nucleases has emerged as a promising platform[7–10]. Several CRISPR-based nucleases such as *Prevotella sp.* P5-125 (PspCas13b) and *Ruminococcus flavefaciens* XPD3002 (RfxCas13d) appear capable of targeting pre-mRNA transcripts with improved specificity and efficiency relative to existing antisense RNA technologies[11–14]. In these systems, the Cas ribonucleoprotein (RNP) is guided to its target RNA transcript by a single CRISPR RNA (crRNA). The crRNA is composed of a direct repeat (DR) stem-loop which mediates RNP formation and a spacer sequence (gRNA) that hybridizes to the target transcript. Cas13 enzymes have two higher eukaryotes and prokaryotes nucleotide-binding (HEPN) domains that mediate RNA cleavage upon target recognition. HEPN-inactive or catalytically

[1]Department of Biomedical Engineering, Duke University, Durham, NC, USA. [2]Department of Surgery, Duke University School of Medicine, Durham, NC, USA. [3]Department of Molecular Genetics & Microbiology, Duke University School of Medicine, Durham, NC, USA. ✉e-mail: aravind.asokan@duke.edu

dead dCas13 uses a distinct ribonuclease activity to process guide RNAs and hence, can serve as an RNA-guided module/effector for targeting specific RNA elements[11,15,16].

Here, we envisioned that the type VI CRISPR-Cas13 system can be repurposed as an effector for trans-splicing, specifically by targeting a recombinant RNA carrying one or more exons to an endogenous pre-spliced transcript. To achieve such, we replaced the antisense binding sequence in the trans-splicing RNA with a type VI CRISPR RNA and provided the cognate CRISPR/Cas13 protein with abolished catalytic activity (representative sequences in Supplementary Table 1). The overall approach, CRISPR Assisted RNA Fragment Trans-splicing (CRAFT) involves co-expression of a recombinant trans-splicing CRAFT RNA fragment (rcRNA) and a modified, cognate type VI CRISPR nuclease. We validate CRAFT across different transformed and primary cell lines, multiple endogenous transcripts, develop a barcoded-based approach for guide selection to facilitate mRNA trans-splicing and affirm that CRAFT can serve as a promising approach to augment spliceosome-mediated trans-splicing.

## Results

### Different Cas13 orthologs can facilitate 5′ and 3′ RNA editing through trans-splicing

Type VI CRISPR guide RNA arrays are expressed on a single RNA molecule and processed by the effector protein[9,10,12]. Type VI-B enzymes process their guide RNA 3′ of the direct repeat structure, while Type VI-A and D enzymes process their guide RNA 5′ of the direct repeat structure. The processing event liberates RNA that is not attached to the direct repeat from the RNP complex. These complementary aspects of Cas13 biology informed the design for the first iteration of CRAFT, in which we employ a PspCas13b and RfxCas13d to facilitate 5′ and 3′ CRAFT respectively. This design consideration enables Cas13 guide RNA processing in such a manner that does not separate the targeting domain (guide RNA) and the exons of the rcRNA into two distinct RNA species.

As proof-of-concept for this RNA editing strategy, we first constructed a split-enhanced green fluorescent protein (EGFP) reporter, where the open reading frame was separated into two halves (splitGFP) by intron 10/11 of the human Lamin A gene (*LMNA*). We then abolished fluorescent expression by introducing a premature stop codon into the 5′ exon of EGFP. To restore EGFP expression, we delivered a two-component CRAFT system to cells containing the reporter. The first component expresses a recombinant CRAFT RNA (rcRNA) composed of the first exon of EGFP followed by a splice donor sequence and a Cas13b crRNA from PspCas13b targeting the *LMNA* intron[10]. The second component expresses a truncated (Δ984-1090), catalytically dead (H133A) PspCas13b enzyme flanked by two nuclear localization signals[10] (Psp-dCas13b) (Fig. 1a). Following transfection of HEK293 cells, Cas protein expression was confirmed by western blot (Supplementary Fig. 1) and EGFP expression measured by flow cytometry as a proxy for mRNA rewriting (Supplementary Fig. 2). Through this approach, referred to herein as 5′ CRAFT, we observed a significant rescue of EGFP expression only when both Cas and an rcRNA containing a guide targeted to the *LMNA* intron 10/11 in the splitGFP reporter were delivered to cells, but background levels with the rcRNA alone or scrambled control (Fig. 1b, c). Thus, 5′ CRAFT can efficiently replace the exons upstream of a target intron in a messenger RNA.

Next, we applied this strategy to replace the 3′ exon using the splitGFP reporter, wherein a stop codon was incorporated into the second exon (Fig. 1d). The 3′ rcRNA was redesigned to contain a guide RNA from RfxCas13d targeting the *LMNA* intron 10/11, followed by a synthetic hemi-intron (branch point, poly-pyrimidine tract, and splice acceptor) and the second exon of splitGFP. The catalytically dead cognate Cas ortholog flanked by two nuclear localization signals was expressed alongside this construct (Rfx-dCas13d)[11,12]. This version of the RNA editing system is referred to herein as 3′ CRAFT. Upon delivering 3′ CRAFT to cells expressing the reporter construct, we again observed significant rescue of EGFP compared to controls (Fig. 1e, f). Thus, CRAFT can also efficiently replace exons in the 3′ regions of a target mRNA transcript. We repeated these proof-of-concept experiments in several common research cell lines: A549 (lung carcinoma), HeLa (cervical carcinoma), and HepG2 (hepatocarcinoma) and observed that CRAFT exhibits comparable efficiencies across diverse cell types (Supplementary Fig. 3). Interestingly, a modest level of trans-splicing was observed in the absence of Cas13 for 3′ but not 5′ CRAFT. While the similarity between antisense targeting sequences of rcRNA and SMaRT trans-splicing RNA can partly account for these observations, mechanistic differences between 3′ and 5′ trans-splicing are unclear.

Further, we demonstrate the ability of CRAFT to support RNA editing by utilizing other trans-splicing-dependent outcomes. For instance, rather than restoring EGFP expression, we show EGFP can be edited to blue fluorescent protein (BFP) by changing the sequence of the exon in the rcRNA[17]. We also demonstrate that CRAFT can be utilized to not only rescue EGFP expression, but re-localize EGFP to the nucleus of cells by attaching a histone 2b (H2B) sequence[18] to the edited transcript (Supplementary Fig. 4). Interestingly, due to the sequence identity retained between the splitGFP reporter and intron 10/11 in the endogenous *LMNA* transcript, we were also able to detect splicing of either rcRNA into the endogenous *LMNA* transcript (Supplementary Fig. 5).

### Proper recruitment of Cas13 is essential for facilitating and augmenting RNA trans-splicing efficiency

To test whether Cas13 is essential to support trans-splicing relative to Watson–Crick base pairing alone, we constructed two additional rcRNAs. The first lacks the Cas13 direct repeat (rcRNA(-DR)), and thus should not form a ribonucleoprotein complex with the Cas13 protein. The second contains a stuffer sequence between the direct repeat and spacer in the CRISPR RNA. This stuffer moves the targeting region (spacer) of the rcRNA out of the central binding channel of Cas13[11,15]. Upon delivery of these modified rcRNAs with Cas13 and the splitGFP reporter to HEK293 cells, we observe attenuated EGFP rescue relative to the original rcRNA (Fig. 1g). Further, we demonstrate the orthogonality of CRAFT by swapping the direct repeat of PspCas13b with the direct repeat from *Porphyromonas Gulae* (PguCas13b). Expression of each Cas protein with an rcRNA containing the cognate direct repeat, rescued EGFP. However, when the rcRNAs were swapped mismatching the protein-RNA pairs, EGFP expression was not rescued (Fig. 1h). As with the 5′CRAFT approach deletion of the direct repeat, or addition of a stuffer sequence between the direct repeat and spacer dramatically attenuated EGFP rescue for 3′CRAFT (Fig. 1i). Swapping of the direct repeat of RfxCas13d that of *Leptotrichia Wadei* (LwaCas13a) decreased rescue of EGFP expression in a similar pattern to that shown by 5′ CRAFT DR swaps (Fig. 1j). Together, these data support the essential role of Cas13 in augmenting trans-splicing. Although further structural and biophysical studies may provide additional mechanistic insight, it is plausible that the ability of Cas13 to stabilize the interaction between the rcRNA and target transcript forms the underlying basis for CRAFT.

### Intron position, guide length, and Cas ortholog choice influence trans-splicing efficiency

To optimize CRAFT, we explored the impact of spacer position along the intron, spacer length, and different Cas orthologs on 5′ and 3′ trans-splicing efficiency using the splitGFP reporter assay. For 5′CRAFT, of the 5 guide sequences tested, the optimal target site was most proximal to the branch point without masking the branch adenosine of the intron (Fig. 2a). We also tested a range of spacer lengths ranging from 30 bp to 150 bp and did not observe a significant increase in GFP intensity beyond 30 bp (Fig. 2b). We then evaluated the compatibility of several Cas13b enzymes with 5′ CRAFT[10,15,19–21] (Fig. 2c). PspCas13b

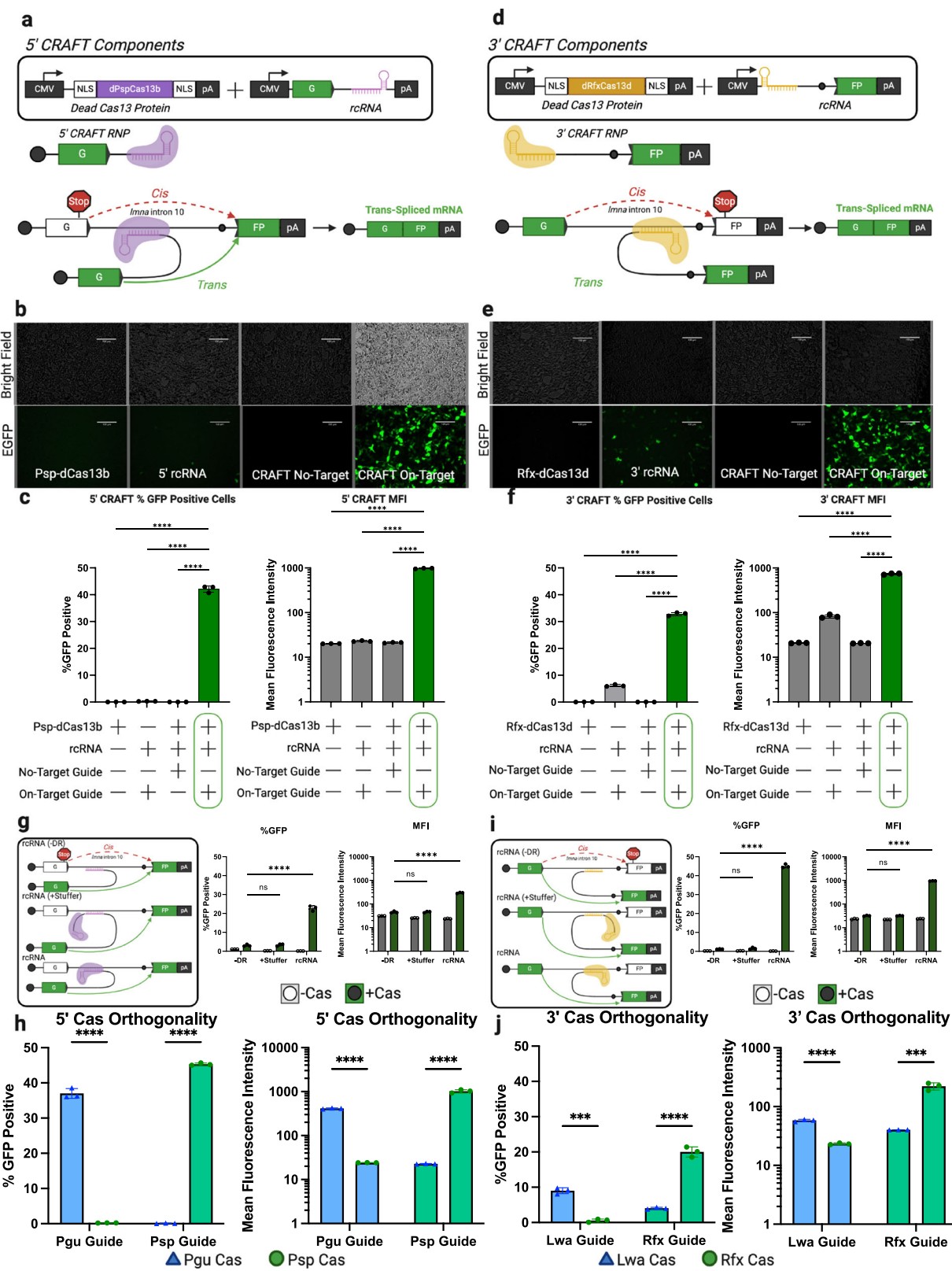

exhibited the highest activity, while PguCas13b performed comparably. Therefore, we recommend the use of PspCas13b and a 30 bp spacer sequence for 5′CRAFT and testing spacers that hybridize to a region 5′ proximal to the branch point in the target intron.

We then tested these parameters for 3′CRAFT. Here, the best spacer targeted a region ~250 bp downstream of the splice donor sequence in the intron (Fig. 2d). Extending the spacer up to 50 bp from

30 bp provided a significant increase in trans-splicing efficiency (Fig. 2e). Further extension of the spacer did not predictably correlate with enhanced trans-splicing, as has been reported for other RNA editing methods[22,23]. It is plausible that different applications for Cas13 may benefit from extended Watson–Crick base pairing as that may improve the affinity for a given target. However, this may come at the detriment of the cleavage activity for applications of catalytically

**Fig. 1 | CRAFT efficiently rescues EGFP expression via precise RNA trans-splicing. a** Schematic of 5′ CRAFT RNA editing. **b** bright field (top) and fluorescent images (bottom) of HEK293 cells transfected with Psp-dCas13b, 5′ rcRNA, Psp-dCas13b and non-targeting 5′rcRNA, or Psp-dCas13b and a on-target 5′rcRNA. **c** Quantitation of flow cytometry for 5′CRAFT targeting *LMNA* intron showing percent GFP positive cells (left) and mean fluorescence intensity (MFI) (right) (Data are mean ± s.d. from *n = 3* individual samples). **d** Schematic of 3′ CRAFT RNA editing. **e** bright field (top) and fluorescent images (bottom) of cells transfected with Rfx-dCas1d, 3′ rcRNA, Rfx-dCas13d and non-targeting 3′rcRNA, or Rfx-dCas13d and on-target 3′rcRNA. **f** Quantitation of flow cytometry for 3′CRAFT targeting *LMNA* intron showing percent GFP positive cells (left) and MFI (right) (Data are mean ± s.d. from *n = 3* individual samples). **g** Schematic diagraming modified 5′rcRNAs (left) that lack the direct repeat (-DR), separate the spacer from the direct repeat (+stuffer), and original rcRNA. Quantitation by flow cytometry for of EGFP rescue

using 5′rcRNA (Data are mean ± s.d. from *n = 3* individual samples). **h** Quantification of EGFP rescue using orthogonal protein/RNA partners in 5′CRAFT; rcRNAs that contain the direct repeat from either PguCas13b (blue bar) or PspCas13b (green bar) along with either dPguCas13b enzyme (blue triangle) or dPspCas13b enzyme (green circle) (Data are mean ± s.d. from *n = 3* individual samples). **i** Schematic diagraming modified 3′rcRNAs. Quantitation by flow cytometry for of EGFP rescue using 3′rcRNA (Data are mean ± s.d. from *n = 3* individual samples). **j** Quantification of EGFP rescue using orthogonal protein/RNA partners in 5′CRAFT; rcRNAs that contain the direct repeat from either PguCas13b (blue bar) or PspCas13b (green bar) along with either LwaCas13d (blue bar) or RfxCas13d (green bar) along with either dLwaCas13d enzyme (blue triangle) or dRfxCas13d enzyme (green circle) (Data are mean ± s.d. from *n = 3* individual samples). Statistical significance was determined by unpaired two-tailed Student *t*-test. *$P < 0.05$; **$P < 0.01$; ***$P < 0.001$; ****$P < 0.0001$; ns not significant. Source data are provided as a Source Data file.

active Cas13 as target recognition and cleavage are dynamically linked[24].

Further, 3′ CRAFT was compatible with several Cas13a and Cas13d orthologs, with RfxCas13d exhibiting the highest efficiency[8,12,21,25] (Fig. 2f). We also examined the use of RfxCas13d and PspCas13b for 5′ and 3′ CRAFT respectively. This architecture of the CRAFT rcRNA did not yield productive trans-splicing (Fig. 2e, f). This may likely be explained by the mechanism through which Cas13 proteins liberate their crRNA from a CRISPR array, justifying our initial design considerations for CRAFT[11,15,16].

## 5′ and 3′ CRAFT can edit mRNA by engaging diverse intronic targets

With these design principles in mind, we set out to demonstrate the versatility of this approach by designing guide RNAs against three additional introns for 5′CRAFT and 3′CRAFT respectively (Fig. 3a, b). The introns were selected from a diverse set of genes underpinning monogenic disease and varied greatly in length, GC content, and species[26]. The first intron is from *RYR2* (intron 95/96), which encodes the ryanodine receptor 2. With an intronic length of 1291 bp and 31% GC content, this represents a prototypical human intron in both size and composition. Following screening of five spacers we observed 48% GFP positive cells for 5′ and 24% for 3′ CRAFT. Next, we targeted a more complex intron from *DMPK* (intron 13/14). This intron is relatively short for a human intron (330 bp) but has an uncharacteristically high GC content. We achieved robust EGFP rescue for both 5′ (20% GFP positive cells) and 3′ CRAFT (30% GFP positive cells) at this target intron. We also show EGFP rescue in splitGFP reporter containing *FXN* intron 1. We achieve EGFP rescue of 24% for 3′ CRAFT at this intron. We also demonstrate the compatibility of CRAFT across species by demonstrating a 5′ CRAFT approach for murine *dmd*. We designed a splitGFP reporter containing intron 23 (2607 bp in length | 31%GC content) of murine *dmd* to identify guides that may allow for rewriting of exons 1–23. We found multiple guides that achieved >33% EGFP rescue when delivered with the reporter and Cas13 to mouse c2c12 myoblasts. Together, CRAFT achieved robust editing across a diverse set of target introns here, highlighting the versatility of this platform.

## 5′ and 3′ CRAFT can efficiently edit endogenous mRNA transcripts

To assess whether CRAFT can edit endogenous mRNA transcripts expressed from a genomic locus, we generate a stable HEK293 cell line expressing the splitGFP reporter using a lentiviral vector system. Briefly, SpyCas9 and corresponding guides were utilized to disrupt the open reading frame in either the 5′ or 3′ exon of stably integrated splitGFP, thereby abolishing expression (Fig. 4a). Reporter cells treated with 5′ or 3′ CRAFT demonstrated robust rescue of EGFP expression at levels comparable to plasmid-based assays (Fig. 4b, d). In addition, we transduced the splitGFP reporter cell line with AAV

vectors encoding 5′ or 3′ CRAFT components. Editing efficiencies ranged between 4 and 9% for 5′ and 3′ CRAFT using AAV vectors, which likely require additional parameter optimization (Fig. 4c, e). Overall, these results confirm that CRAFT can edit endogenous mRNA transcripts and is amenable to delivery by AAV vectors.

To determine the absolute trans-splicing efficiency of CRAFT, we replaced the fragments of EGFP in either rcRNA with either exons 1–10 or exons 11–12 of *LMNA* in the 5′ and 3′ rcRNA respectively. To differentiate cis- and trans-spliced transcripts, we included a silent mutation in exon 10 of the 5′ rcRNA construct and 3xFlag tag on the n-terminus of the open reading frame (Fig. 4f). Similarly, we included a silent mutation in exon 11 of the 3′ construct and we appended a 3xFlag tag to the c-terminus of the open reading frame as well as a synthetic 3′ UTR containing the polyadenylation sequence from simian virus 40 (SV40) (Fig. 4g). Through this process of replacing a fragment of an mRNA, we were able to validate editing of *LMNA* mRNA through Sanger sequencing (Fig. 4h, j). We analyzed trans-splicing efficiency by targeted amplicon sequencing of the *LMNA* mRNA across the exon 10/11 splice junction to determine the percent of reads that contain the silent mutation. Briefly, a primer set was designed to anneal to exons 10 and 11 encompassing the locations of each silent mutation for the respective strategies. Cis- and trans-spliced transcripts should be amplified at equal efficiency by these primers because the binding sites are identical and the amplicons only vary by a single nucleotide if editing occurs. Thus, the percent of reads containing the cytidine to guanosine (C > G) mutation in exon 10 corresponds to 5′ CRAFT trans-splicing efficiency. While the percent of reads containing the silent thymine (uridine in RNA transcript) to cytidine (T > C) mutation in exon 11 corresponds to 3′ CRAFT trans-splicing efficiency (Supplementary Fig. 6). We observed 25.13% editing of the endogenous *LMNA* transcripts through the 5′ CRAFT approach and 22.60% editing with 3′ CRAFT (Fig. 4i, k).

To further assess functional impact of editing the *LMNA* transcript, we appended a FLAG tag to either the N- or C- termini of Lamin A using CRAFT and confirmed colocalization of endogenous Lamin A and the FLAG-tagged protein by immunofluorescence (Fig. 4l, m). Additionally, we demonstrate how CRAFT can be used to decouple the impact of genetic mutations at the DNA and RNA level, examining the genetics underpinning Hutchinson−Gilford progeria (HGPS). HGPS is a rare autosomal dominant disorder, that is caused by a mutation in exon 11 of *LMNA*. This mutation activates a cryptic splice site and makes an RNA that encodes a toxic protein product called progerin. Correction of this mutation at the genome level using base editing has been demonstrated to reduce progerin without affecting *LMNA* mRNA in patient-derived fibroblasts and a mouse model of HGPS[27]. Here, we show that 3′ CRAFT-mediated correction of this mutation at the mRNA level effectively decreased progerin mRNA by ~47% in a patient fibroblast cell line without decreasing *LMNA* transcript expression, corroborating the effect shown at the DNA level (Supplementary Fig. 7).

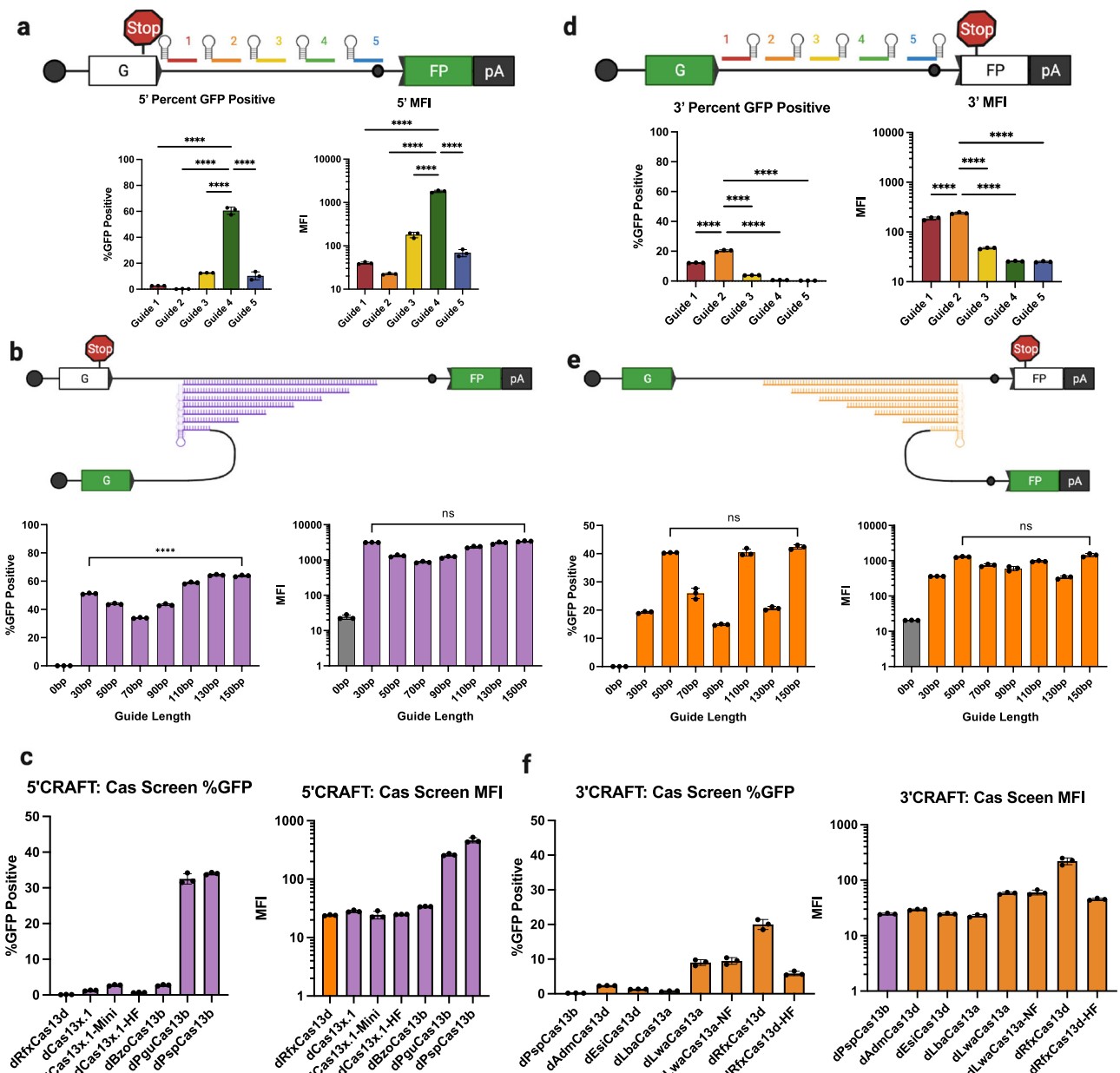

**Fig. 2 | Screen of parameters that affect CRAFT-mediated RNA editing.**
**a** Schematic guide tiling for 5′CRAFT rcRNAs along the target intron (top). Position and color of guide correlates with the quantification by flow cytometry presented as percent GFP positive cells and MFI (below) (Data are mean ± s.d. from $n = 3$ individual samples). **b** Lead guide candidate from the (**a**) (guide 4) was modified by extending the original 30 bp guide to 150 bp in 20bp step increments and evaluated in the splitGFP reporter assay. Quantitation by flow cytometry for the 5′CRAFT guide length: percent GFP positive cells (left) and MFI (right) (Data are mean ± s.d. from $n = 3$ individual samples). **c** The direct repeat in the rcRNA of the lead guide candidate from (**a**) was swapped with the direct repeat from alternative type VI CRISPR species and co-transfected with the cognate catalytically dead cas13 protein. Quantitation by flow cytometry is presented as percent GFP positive cells (left) and MFI (right) (Data are mean ± s.d. from $n = 3$ individual samples). **d** Schematic guide tiling for 3′CRAFT rcRNAs along the target intron (top). Position and color of

guide correlates with the quantification by flow cytometry presented as percent GFP positive cells and MFI (below) (Data are mean ± s.d. from $n = 3$ individual samples). **e** Lead guide candidate from (**d**) (guide 2) was modified by extending the original 30 bp guide to 150 bp in 20 bp step increments and evaluated in the splitGFP reporter assay. Quantitation by flow cytometry for the 3′CRAFT guide length: percent GFP positive cells (left) and MFI (right) (Data are mean ± s.d. from $n = 3$ individual samples). **f** The direct repeat in the 3′rcRNA of the lead guide candidate of *LMNA* (guide 2) was swapped with the direct repeat from alternative type VI CRISPR species and co-transfected with the cognate catalytically dead cas13 protein. Quantitation by flow cytometry is presented as percent GFP positive cells (left) and MFI (right) (Data are mean ± s.d. from $n = 3$ individual samples). Statistical significance was determined by One-Way ANOVA with Tukey's post-test. *$P < 0.05$; **$P < 0.01$; ***$P < 0.001$; ****$P < 0.0001$; ns not significant. Source data are provided as a Source Data file.

## Off-target characterization of CRAFT

We performed bulk RNA sequencing to assess unintended transcriptional changes induced by CRAFT. When compared to a non-targeting guide sequence we saw minimal alterations in gene expression (Fig. 4n, o). Since CRAFT rewrites RNA rather than knocking down expression, assaying gene expression is not sufficient to characterize

alterations to the transcriptome. To directly assess off-target trans-splicing into RNAs other than the endogenous *LMNA* transcript, we used Arriba, a STAR aligner tool, which performs unbiased detection of gene fusions from bulk RNA sequencing data[28]. Arriba did not return any fusions containing the exons delivered in the rcRNA for either 5′ or 3′ CRAFT (Source Data). Further, we attempted to increase the

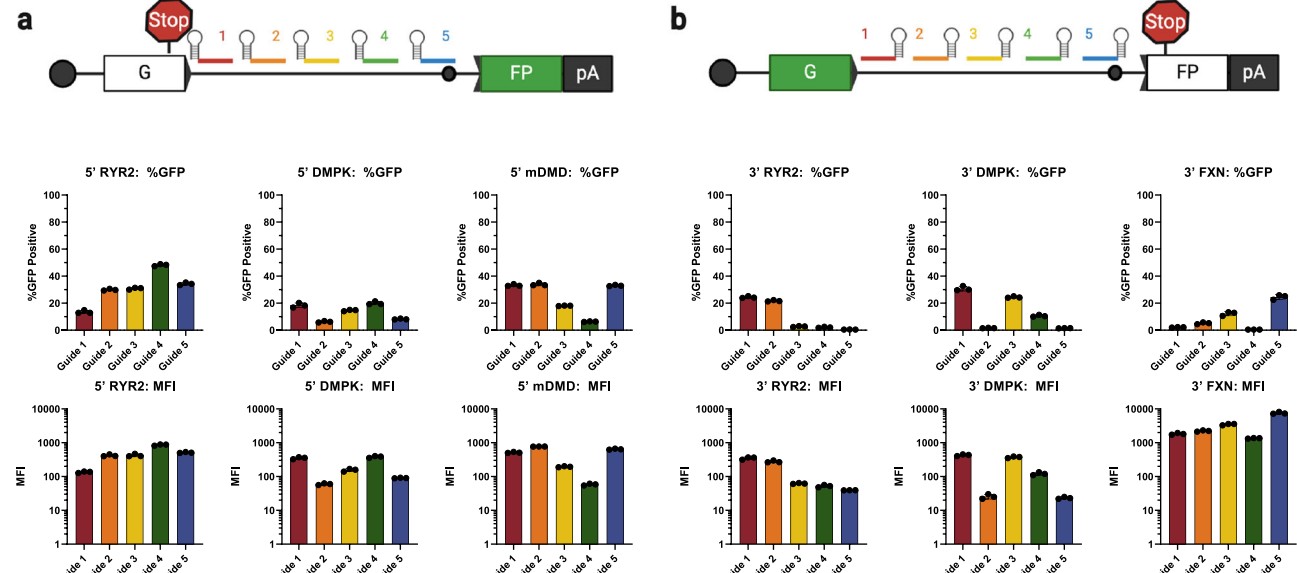

**Fig. 3 | CRAFT enables efficient rewriting across a diverse set of target introns.**
**a** Schematic guide tiling for 5'CRAFT rcRNAs along the target intron (top). Target position and color of guide target sequence correlates with the quantification by flow cytometry presented as percent GFP positive cells and MFI (below) across three different introns: human *RYR2* intron 95/96, human *DMPK* intron 13/14, mouse *dmd* intron 23/24 (Data are mean ± s.d. from *n = 3* individual samples). **b** Schematic guide tiling for 3'CRAFT rcRNAs along the target intron (top). Target position and color of guide target sequence correlates with the quantification by flow cytometry presented as percent GFP positive cells and MFI (below) across three different introns: human *RYR2* intron 95/96, human *DMPK* intron 13/14, human *FXN* intron 1/2 (Data are mean ± s.d. from *n = 3* individual samples).

sensitivity of our off-target analysis by enriching for transcripts containing the exons included in the rcRNA and performed Nanopore-based long read sequencing[11,15,16,29]. However, this enrichment provided us only with reads that mapped to rcRNAs that were not spliced. Factors such as read depth, transcript abundance, or limit of detection may account for these observations.

**5' and 3' CRAFT efficiency exceeds other trans-splicing methods**
The design of our CRAFT platform was inspired by spliceosome-mediated RNA trans-splicing (SMaRT). The salient difference between the two platforms is the mechanism of targeting, while SMaRT utilizes an antisense RNA sequence to complex with a target intron by Watson–Crick base pairing alone; CRAFT exchanges antisense binding for a Cas13 guide RNA complex, which directs target engagement to improve trans-splicing efficiency. To test this hypothesis, we removed the direct repeat from all top-performing rcRNAs to make analogous SMaRT pre-mRNA trans-splicing molecules (PTMs). We compared each platforms capacity to rescue EGFP expression and observed a marked improvement with CRAFT over SMaRT across all matched guide sequences (Fig. 5a). Specifically, we observed ~5-to-40 fold increase in trans-splicing efficiency across both 5' and 3' target introns derived from *LMNA*, *RYR2*, *DMPK*, and *dmd* (murine) pre-mRNA transcripts. While the rules surrounding guide design may be different, we hypothesized that CRAFT can be used to enhance previously engineered SMaRT designs without additional spacer selection. Specifically, we want to highlight that one of the most robust examples of RNA rewriting by SMaRT was shown through the replacement of the 3' exons of *RHO* at an efficiency of >40% in a plasmid reporter[30]. We hypothesized that by incorporating the same guide sequence into a 3'CRAFT strategy, we could improve trans-splicing efficiency. Though this comparison of the two platforms with the same binding domain shows that CRAFT rescues EGFP expression nearly three times more effectively than SMaRT (Fig. 5a, b). It is noteworthy to mention that despite utilizing the same target intron and binding domain, albeit with a different reporter, we were unable to achieve greater than 7-8% editing

efficiency using SMaRT in this study. To more robustly evaluate SMaRT vs CRAFT we then designed three guides (without optimization) for either approach targeting the same intronic region of the human dystrophin (*DMD*) transcript. Consistent with earlier observations, we observed enhanced trans-splicing efficiency with CRAFT relative to SMaRT when targeting the endogenous human *DMD* transcript encoding the *dp71* isoform in vitro (Fig. 5c–e). Taken together, these results, along with the multiple intronic targets in the context of GFP (Fig. 5a) we feel that substantial evidence is in place to corroborate that CRAFT can improve trans-splicing efficiency.

Separately, during the writing of this manuscript, a similar Cas13-based RNA trans-splicing method (Splice Editing) was reported[31]. While CRAFT features a single trans-splicing RNA containing crRNA, hemi-intron, and exon(s), Splice Editing separates targeting and trans-splicing functionality across two RNA species. Briefly, a Cas13 crRNA mediates targeting, while a second trans-splicing RNA (repRNA) replaces the targeting domain with MS2 stem-loops (repRNA). These two RNAs are expressed with an MS2 coat protein c-terminally fused to Psp-dCas13b (Psp-dCas13b-MS2). The crRNA complexes with the Cas13 protein and targets the RNP to the target intron, while the repRNA leverages the MS2 coat protein-stem loop interaction to form the fully assembled RNP. We compared CRAFT and Splice Editing with guides that we had optimized for Psp- and Rfx-Cas13. Although both approaches share the principle of Cas13-mediated RNA targeting, CRAFT outperformed Splice Editing across several architectures of Splice Edit in both 5' and 3' exon replacement contexts (Supplementary Fig. 8).

**Towards high-throughput CRAFT design**
The aforementioned experiments with *LMNA* and *DMD* as a target helped establish the principle that single-nucleotide polymorphisms (SNPs) within edited RNA transcripts can serve as a measure of trans-splicing efficiency. As a step towards expanding the adaptability of CRAFT as an RNA editing tool, we exploited this observation to design a high-throughput screen comprised of a library of guide sequences tiling the *LMNA* intron 10/11. Briefly, we generated a pool of oligos that

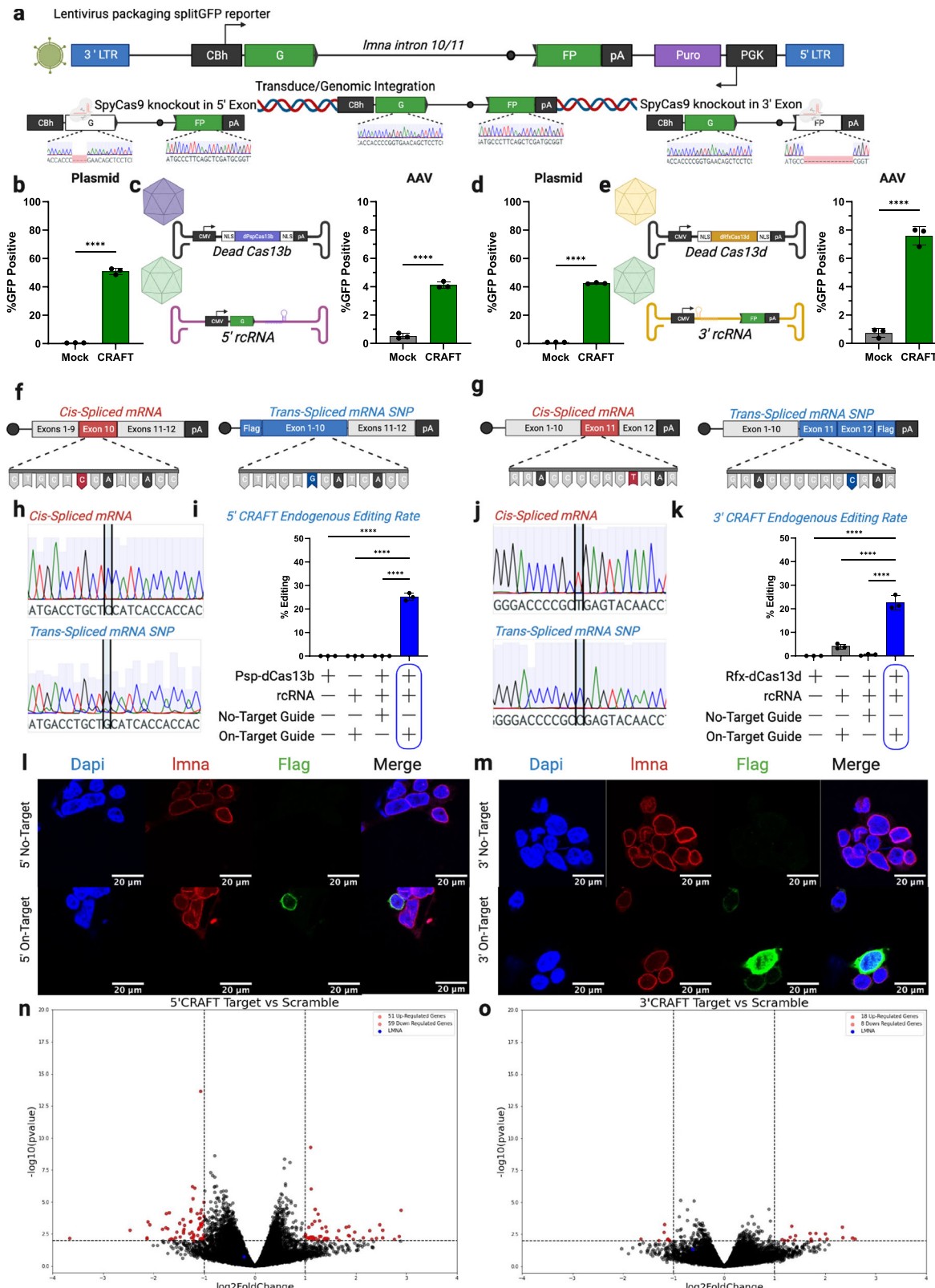

contained a spacer sequence, hemi-intron, and beginning of exon 11. Each oligo in the pool contained a unique spacer that was operably linked to a "barcode" of SNPs in exon 11. This oligo pool was then cloned into the rcRNA expression plasmid between the direct repeat and remaining exon 11 sequence. This plasmid pool was transfected along with Rfx-dCas13d to HEK293 cells. Three days later, we performed the same targeted amplicon sequencing on the RNA from

these cells as was done for measuring the efficiency of CRAFT at endogenous targets (Fig. 6a). Thus, by knowing the barcode of a guide and sequencing across the exon 10/11 splice junction in the mature transcript we were able to determine the relative efficiency of each guide by calculating the abundance of its associated barcode in the mRNA and comparing that to its abundance in the plasmid pool delivered to the cells (Fig. 6b). The trend of relative efficiencies

**Fig. 4 | CRAFT is amenable to viral delivery and rewriting of endogenous mRNA. a** Schematic of stable HEK293 splitGFP reporter cell line generation accompanied by Sanger sequencing traces. **b** EGFP rescue in HEK293 cells that stably express the splitGFP reporter following transfection of 5′CRAFT plotted as percent GFP positive cells. **c** EGFP rescue in HEK293 cells that stably express the splitGFP reporter following transduction with 5′CRAFT (AAV genome schematics shown left) and plotted as percent GFP positive (right). **d** EGFP rescue in HEK293 cells that stably express the splitGFP reporter following transfection of 3′ CRAFT plotted as percent GFP positive cells. **e** EGFP rescue in HEK293 cells that stably express the splitGFP reporter following transduction with 3′CRAFT (AAV genome schematics shown left) and plotted as percent GFP positive cells (right). **f** Schematic of cis- (left) trans- (right) endogenous *LMNA* transcripts highlighting a silent (C > G) snp in the trans-spliced. **g** Schematic of cis- (left) trans- (right) endogenous *LMNA* transcripts highlighting a silent (T > C) snp in the trans-spliced. **h** Sanger sequencing of the cis- (top) and trans- (bottom) spliced *LMNA* transcripts.

**i** frequency of 5′CRAFT editing in endogenous *LMNA* transcripts. **j** Sanger sequencing of the cis- (top) and trans- (bottom) spliced *LMNA* transcripts. **k** frequency of 3′CRAFT editing in endogenous *LMNA* transcripts. **l** confocal microscopy images of N-terminal Flag tag introduced by 5′CRAFT Dapi (nuclear staining), AlexaFluor594 (Lamin A/C), and Flag (AlexaFluor488) (representative images from *n = 2* independent experiments). **m** confocal microscopy images of C-terminal Flag tag introduced by 3′CRAFT (staining panel same as l). **n** volcano plot generated by DEseq2 analysis of differential gene expression from bulk RNA-seq in HEK293 cells transfected with dPspCas13b and either a targeting or non-targeting guide in the 5′ rcRNA. **o** volcano plot generated by DEseq2 analysis of differential gene expression from bulk RNA-seq in HEK293 cells transfected with dRfxCas13d and either a targeting or non-targeting guide in the 3′ rcRNA. All data are mean from *n = 3* individual samples. Statistical significance was determined by unpaired two-tailed Student *t*-test. *P < 0.05; **P < 0.01; ***P < 0.001; ****P < 0.0001; ns not significant. Source data are provided as a Source Data file.

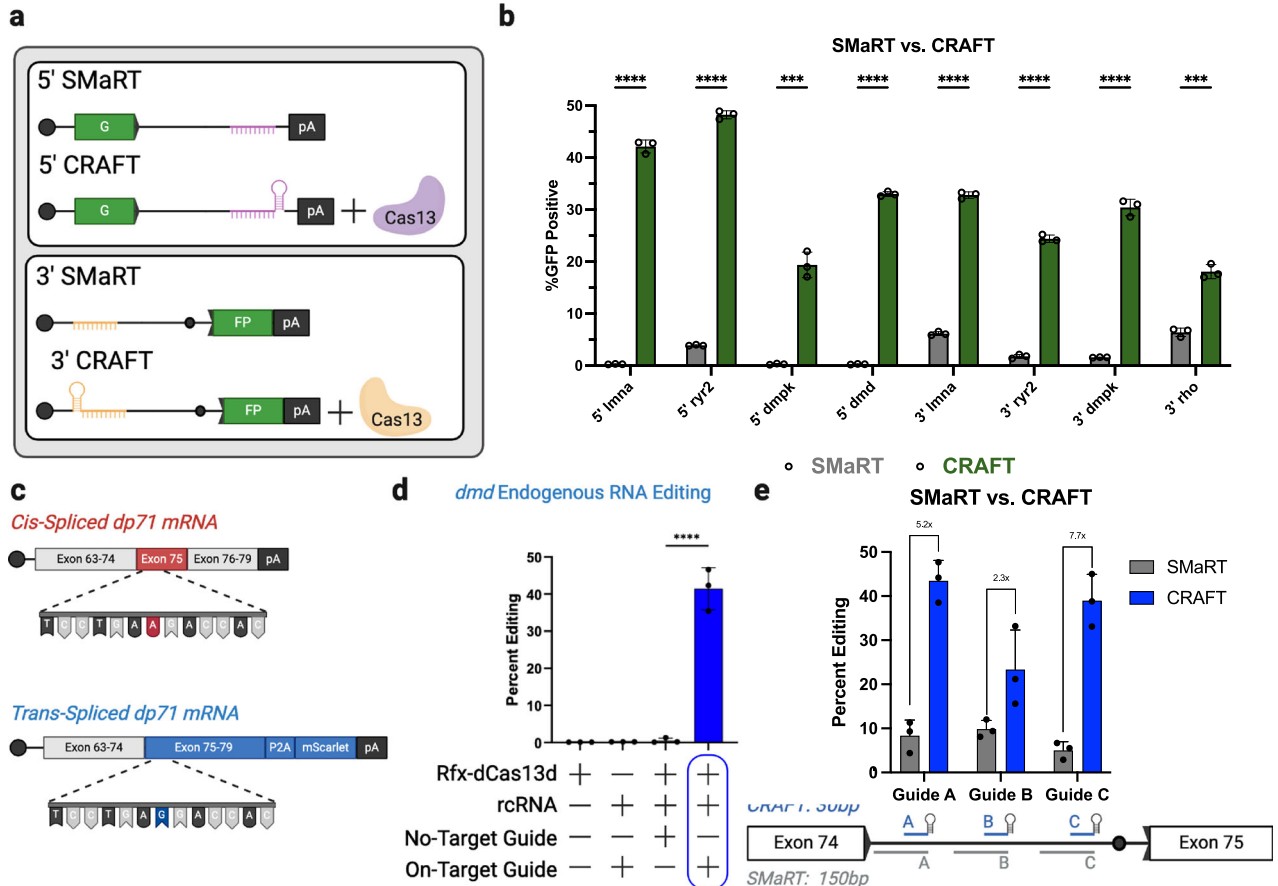

**Fig. 5 | Comparison of SMaRT vs. CRAFT across multiple targets. a** Schematic 5′ SMaRT and CRAFT designs (top). Schematic 3′ SMaRT and CRAFT designs (bottom). **b** Trans-splicing efficiency was quantified as percent GFP positive cells measured by flow cytometry (below) at 5 different introns: *LMNA* intron 10/11 (5′ and 3′ *p* < .0001), *RYR2* intron 95/96 (5′ and 3′ *p* < .0001), *DMPK* intron 12/13 (3′ *p* < .0001), mouse *dmd* intron 23/24 (5′ *p* < .0001), and *RHO* intron 1/2 (3′ *p* < .0001) (Data are mean ± s.d. from *n = 3* individual samples). **c** Schematic of cis-spliced endogenous *DMD* transcript (top) and trans-spliced transcripts (bottom). The position of the single nucleotide polymorphism (A > G) is highlighted. **d** Frequency of 3′CRAFT editing in endogenous *DMD* transcripts. The y-axis is the percent of reads from targeted amplicon sequencing containing the snp mutation. The x-axis

refers to specific treatment: dCas13 alone, rcRNA alone, dCas13 with an rcRNA that does not target the *DMD* intron, and dCas13 with an rcRNA that targets the *DMD* intron (Data are mean ± s.d. from *n = 3* individual samples). **e** Comparison of CRAFT and SmaRT at the *DMD* locus. The y-axis is the percent of reads from targeted amplicon sequencing containing the A > G mutation. Guides targeting the same intronic location for SMaRT (gray) and CRAFT (blue) are plotted on the x-axis (Data are mean ± s.d. from *n = 3* individual samples) Guide A (*p* = .0005), Guide B (*p* = .064), Guide C (*p* = .0007). Statistical significance was determined by unpaired students two-tailed *t*-test. *P < 0.05; **P < 0.01; ***P < 0.001; ****P < 0.0001; ns not significant. Source data are provided as a Source Data file.

corroborated our low-throughput screen using the splitGFP reporter (Fig. 2d). We then applied this library technique to rewrite the last exon of dystrophia myotonica protein kinase (*DMPK*) in HEK293 cells using CRAFT. We found an optimal guide that achieved efficient trans-splicing exceeding 24% (Fig. 6c–e). Notably, this observation (Fig. 6e)

further reiterates our earlier conclusions regarding the potential to achieve improvement over a previously published SMaRT approach to replace the same terminal exon of the *DMPK* transcript (which reported ~7% trans-splicing efficiency)[32]. Interestingly, without the expression of Cas we did not detect significant RNA trans-splicing in the

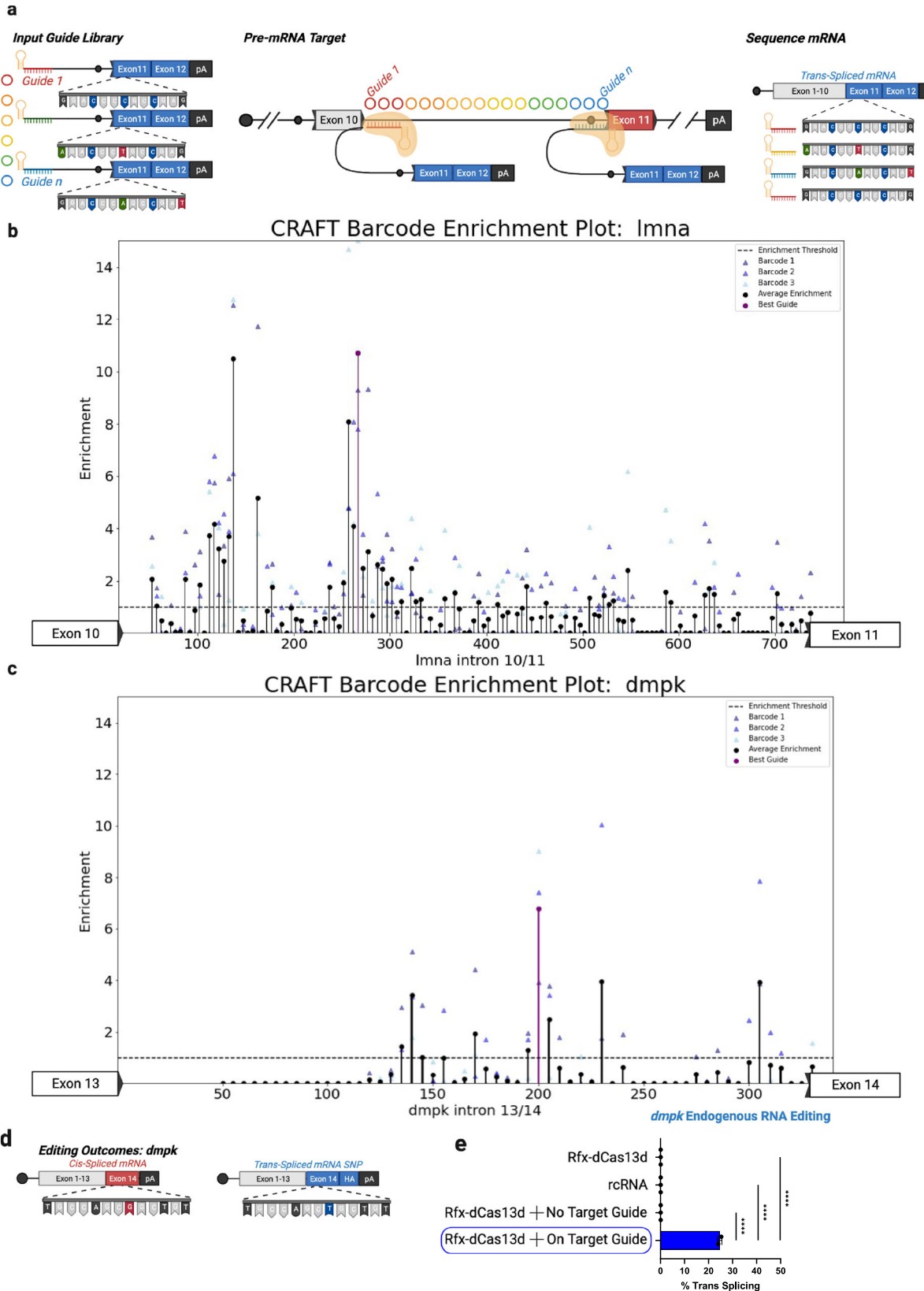

**a** Input Guide Library — Pre-mRNA Target — Sequence mRNA

**b** CRAFT Barcode Enrichment Plot: lmna

**c** CRAFT Barcode Enrichment Plot: dmpk

**d** Editing Outcomes: dmpk

**e** dmpk Endogenous RNA Editing

*DMPK* RNA, corroborating the markedly improved trans-splicing efficiency enabled by an effector protein.

## Discussion

RNA-guided approaches to edit RNA are promising orthogonal platforms to genome editing. Existing mechanisms for programable RNA manipulation include deaminases (A > I or C > U), exon skipping/inclusion, and tunable RNA expression (stabilization or knockdown). Each of these are context-dependent, limited in the range of potential editing outcomes, and require significant redesign for each mutation even within the same target transcript. Thus, RNA trans-splicing is an attractive approach to nucleic acid manipulation, as only a single

**Fig. 6 | High-throughput guide selection through guide coupled barcode.**
**a** Schematic of barcode approach. Briefly, a library of unique barcodes corresponding a specific spacer sequence was delivered to HEK293 cells with Rfx-dCas13d. Functional RNPs target to intron 10/11 of *LMNA* and undergo transsplicing. RNA is harvested and the abundance of each barcode at the start of exon 11 was measured by targeted amplicon sequencing. This barcode corresponds to the guide associated with its trans-splicing. **b** enrichment plot for each guide targeting *LMNA* intron 10/11 as a function of position. Y-axis is barcode enrichment calculated by the (%barcode in trans-spliced RNA / %barcode in input plasmid DNA), and x-axis is the position along *LMNA* intron 10/11 of each guide. Each unique barcode (3) for a single guide is plotted as a triangle and the average of these enrichment scores is plotted as a black circle. The best guide is plotted in purple. **c** enrichment plot for

each guide targeting *DMPK* intron 13/14 as a function of position. **d** Schematic of endogenous *DMPK* transcript (left) and edited *DMPK* transcript (right). Notably, there is a single G > T transition mutation installed between the edited transcript and the endogenous transcript. **e** Frequency of 3′CRAFT editing in endogenous *DMPK* transcripts. The y-axis is the percent of reads from targeted amplicon sequencing containing the snp mutation. The x-axis refers to specific treatment: dCas13 alone, rcRNA alone, dCas13 with an rcRNA that does not target the *DMPK* intron, and dCas13 with an rcRNA that targets the given intron (Data are mean ± s.d. from $n = 3$ individual samples). Statistical significance was determined by unpaired students two-tailed *t*-test. *$P < 0.05$; **$P < 0.01$; ***$P < 0.001$; ****$P < 0.0001$; ns not significant. Source data are provided as a Source Data file.

editing reaction enables programmable rewriting of multiple kilobases in the primary RNA sequence. Within this framework, CRISPR-Cas13-based systems have been extensively evaluated by multiple groups. Here, we repurpose Cas13 to facilitate mRNA trans-splicing and demonstrate the potential to replace large stretches of endogenous transcripts. One of the core findings of this work is that Cas13 enhances RNA trans-splicing in a guide-dependent manner. Although further structural insights are required, it is tempting to speculate that Cas13 functions in this context by stabilizing the RNA-RNA duplex between the spacer and target transcript, likely within the binding groove[11,15,16]. Moreover, a deeper understanding of the interplay between CRAFT and spliceosomal machinery could not only provide mechanistic insight, but also help improve trans-splicing efficiency in general. Nevertheless, we were able to reproduce this finding across multiple Cas orthologs for 5′ and 3′ CRAFT. We anticipate that other emerging Cas orthologs discovered through data mining efforts and other eukaryotic RNA binding proteins can potentially be repurposed for enhanced trans-splicing[13,33–37].

We also developed a pooled screening method for rapid rcRNA guide optimization. Adapting this approach into massively parallel Cas13 guide screens may help generation of robust datasets to train machine learning algorithms[24,35,38]. Further, while this first attempt at a high-throughput barcoded screen investigated a single parameter (guide targeting position), one can envision multiparametric screening strategies that may not only help improve CRAFT, but also identify strategies for trans-splicing methods that do not require addition of exogenous RNA binding proteins. For instance, we demonstrate improved trans-splicing efficiency of CRAFT relative to SMaRT across multiple introns for both 5′ and 3′ exon replacement. We also identified a previously optimized SMaRT RNA and demonstrate that simply appending an RfxCas13d DR upstream of the RNA binding sequence and co-expressing Cas13 enhances trans-splicing efficiency by ~3-fold. These results were corroborated by evaluating direct comparisons of CRAFT and SMaRT across 3 endogenous RNA transcripts in 5′ and 3′ orientations containing spacers of 30 or 150 bp in length. Extending our screening approach may help identify parameters for improvement of SMaRT as well in the future.

Although our preliminary data supporting CRAFT are promising, therapeutic application of this platform faces significant challenges. Notably, the size of Cas13 precludes the development of single AAV vector systems for trans-splicing with this approach. Further, the obligatory long-term expression of Cas13 proteins in different tissues raises concerns regarding potential immunotoxicity and pose the risk of off-target effects as applicable to all other CRISPR-based platforms[9,39–41]. Here, we attempted to address off-target through bulk transcriptome sequencing and targeted sequencing[42,43]. From bulk sequencing we observed minimal differential gene expression following CRAFT editing and no detectable fusion transcripts involving the exons of the rcRNA following best practices analyzing chimeric RNA species in cancer[28,44]. It is possible that improving read depth from bulk RNA sequencing pipelines may help detect rare off-target fusion transcripts. While we were unable to compensate for this limitation in

the current study, despite the use of target enrichment and long-read sequencing, emerging sequencing and bioinformatics pipelines may help in this regard[29,38,44].

Despite these limitations, we integrated CRAFT with recombinant AAV vectors to evaluate potential for therapeutic applications. First, we observed decreased trans-splicing efficiencies of endogenous transcripts when applying CRAFT in vitro using a dual, recombinant AAV vector system. Further, we tested CRAFT using dual AAV vectors in vivo in the *mdx* model of Duchenne Muscular Dystrophy (DMD)[45]. Briefly, in attempting to rewrite exons 1–23 of murine *dmd*, we observed dystrophin-positive muscle fibers and trans-spliced RNA products; however, statistical significance was not achieved. (Supplementary Fig. 9). These preliminary results suggest that CRAFT will require significant optimization prior to preclinical development. Nonetheless, our overall results convincingly demonstrate the potential utility of CRAFT as a tool to facilitate RNA editing through transsplicing. Our data highlights the ability to employ CRAFT to (a) install mutations in RNA transcripts that restore protein expression, (b) modify protein function, and (c) tag endogenous proteins. Moreover, CRAFT may also be used to model disease biology by incorporating 5′ and 3′ edits to RNA transcripts that may affect function or protein expression. Overall, the versatility and efficiency of CRAFT offer exciting prospects as a tool for interrogating cellular RNA and may provide a foundation for therapeutic RNA editing platforms.

## Methods
All research described in the current study is compliant with ethical regulations and approved by the Duke Institutional Biosafety Committee (IBC) and conducted in compliance with the National Institutes of Health (NIH) Guidelines for Research Involving Recombinant or Synthetic Nucleic Acid Molecules (NIH Guidelines). Experiments involving animals were conducted with strict adherence to the guidelines for the care and use of laboratory animals of the National Institutes of Health (NIH). All experiments were approved by the Institutional Animal Care and Use Committee (IACUC) at Duke University.

### Cell culture
HEK293 (ATCC), A549 (ATCC), HeLa (ATCC), and HepG2 (ATCC) cells were maintained in Dulbecco's modified Eagle's medium (Thermo Fisher Scientific) supplemented with 10% fetal bovine serum (Cytiva) and 1% penicillin-streptomycin (Thermo Fisher Scientific) in 5% $CO_2$ at 37 °C. C2C12 (ATCC) mouse myoblasts and HGPS patient-derived fibroblasts (Coriell) were maintained in were maintained in Dulbecco's modified Eagle's medium (Thermo Fisher Scientific) supplemented with 20% fetal bovine serum (Cytiva) and 1% penicillin-streptomycin (Thermo Fisher Scientific) in 5% $CO_2$ at 37 °C. All cells were regularly passaged with 0.05% Trypsin-EDTA (Thermo Fisher Scientific).

### Molecular cloning
rcRNA expression plasmids containing Cas13 direct repeat, Esp3i cut sites, the hemi-intron, exons(s) were ordered from Twist Biosciences.

Guide RNA spacers were cloned into rcRNA vectors by digesting plasmids with Esp3i restriction enzyme (New England Biolabs) for 1 h at 37 °C according to manufacturer's directions. Primers containing the spacer sequence and homology to the rcRNA plasmid backbone were annealed using Q5® High-Fidelity DNA Polymerase (New England Biolabs) for 35-cycles. Digested backbone plasmid and annealed spacer sequences were assembled using NEB High-Fidelity DNA Assembly 2× Master Mix (New England Biolabs) in a reaction containing 5 µl of the master mix 2 µl of the digested backbone and 3 µl of the PCR annealed spacer. This reaction was incubated at 1 h at 50 °C and immediately transformed into DH5α chemically competent E. Coli (representative rcRNA sequences are listed in Supplementary Table 1).

The open reading frame of catalytically dead *Prevotella sp. P5-125* (PspCas13b) and *Ruminococcus flavefaciens* XPD3002 (RfxCas13d) flanked by two nuclear localization signals was ordered from Twist Biosciences and cloned into a CMV expression cassette (specific Cas protein sequences are listed in Supplementary Table 1).

For the splitGFP reporter cloning, either half of the EGFP open reading frame was amplified from pLHA-TR-EF1α-Hipk3-Polio-SplitGFP[46] with primers 49 and 50 (first half of EGFP ORF) and primers 51 and 52 (second half of EGFP ORF) using Q5® High-Fidelity DNA Polymerase (New England Biolabs) for 35-cycles. And cloned into a expression plasmid under the control of the EF1α using NEB High-Fidelity DNA Assembly 2× Master Mix (New England Biolabs). To generate 3′ and 5′ stop codon reporters, stop codons were inserted by site-directed mutagenesis of the splitGFP reporter. The splitGFP plasmid was amplified using primers 53 and 54 to install 3′ stop codon and primers 55 and 56 to install the 5′ stop codon using Q5® High-Fidelity DNA Polymerase (New England Biolabs) for 35-cycles. Plasmids were then assembled in a kinase, ligase, and DpnI reaction (representative splitGFP sequences are listed in Supplementary Table 1). Finally, using the NheI restriction site located between the middle of the EGFP open reading frames, we cloned in the target introns to the splitGFP reporter.

## Transfections

HEK293 cells were seeded at 180,000 cells per well in a 24-well tissue culture plate (VWR) containing 500 µl cell culture media 24 h prior to transfection. For endogenous RNA trans-splicing experiments 400 ng of dCas13 expression plasmid was mixed with 400 ng of rcRNA expression plasmid in 50 µl Dulbecco's modified Eagle's medium (Thermo Fisher Scientific). In a second tube, 3 µl of polyethylenimine (Polysciences) was diluted in 50 µl Dulbecco's modified Eagle's medium (Thermo Fisher Scientific). Tubes one was then added to tube two, vortexed briefly, and incubated at room temperature for 5 min before being added dropwise to cells. For experiments where only dCas13 expression plasmid or rcRNA plasmid was transfected, total DNA was normalized to 800 ng using an EGFP expression plasmid as a transfection control.

For splitGFP reporter experiments, 300 ng of dCas13 expression plasmid was mixed with 300 ng of rcRNA expression plasmid and 300 ng of splitGFP reporter plasmid in 50 µl Dulbecco's modified Eagle's medium (Thermo Fisher Scientific). In a second tube 3.2 µl of polyethylenimine (Polysciences) was diluted in 50 µl Dulbecco's modified Eagle's medium (Thermo Fisher Scientific). Tubes one was then added to tube two, vortexed briefly, and incubated at room temperature for 5 min before being added dropwise to cells. For experiments where only dCas13 expression plasmid or rcRNA plasmid was transfected, total DNA was normalized to 900 ng using pcDNA3.1 plasmid.

A549, HeLa, HepG2, C2C12, and HGPS patient-derived cells were transfected using the Neon Electroporation System (Thermo Fisher Scientific) according to manufacturers instructions and seeded in a 24-well tissue culture plate (VWR) containing 500 µl cell culture media

following transfection. For endogenous RNA trans-splicing experiments 400 ng of dCas13 expression plasmid was mixed with 400 ng of rcRNA expression plasmid 10 µl R Buffer (Thermo Fisher Scientific). For experiments where only dCas13 expression plasmid or rcRNA plasmid was transfected, total DNA was normalized to 800 ng using an EGFP expression plasmid as a transfection control. For splitGFP reporter experiments, 300 ng of dCas13 expression plasmid was mixed with 300 ng of rcRNA expression plasmid and 300 ng of splitGFP reporter plasmid. For experiments where only dCas13 expression plasmid or rcRNA plasmid was transfected, total DNA was normalized to 900 ng using pUC19 plasmid.

## RNA extraction

Cells were harvested at 72 h post-transfection by aspirating the media and dissociating cells in 500 µl phosphate-buffered saline (Thermo Fisher Scientific). Cells were then centrifuged at $300 \times g$ for 5 min and the supernatant was aspirated. Following cell harvest, all RNA extractions were performed with TRIzol reagent according to manufacturer's directions.

## Validation of trans-splicing by Sanger sequencing

Purified RNA was reverse transcribed to cDNA using High-Capacity RNA-to-cDNA Kit (Thermo Fisher Scientific) according to manufacturer's directions. Target cDNA were then amplified using one primer that annealed specifically to the target RNA and a second primer that annealed specifically to the rcRNA (primers 45/46 for *DMD* and primers 47/48 for *LMNA*) with Q5® High-Fidelity DNA Polymerase (New England Biolabs) for 35-cycles. PCR reactions cleaned up by electrophoresis on a 1% agarose gel and extracted using a Gel Extraction and PCR cleanup Kit (IBI Scientific). Purified DNA was then submitted for Sanger sequencing (Genewiz). All primers used for Sanger sequencing are reported in Supplementary Table 2.

## Library preparation for targeted amplicon sequencing

Purified RNA was reverse transcribed to cDNA using High-Capacity RNA-to-cDNA Kit (Thermo Fisher Scientific) according to manufacturer's directions. Target cDNA were then amplified using primers that span the trans-splice junction (primers 41/42 for *DMD* and primers 43/44 for *LMNA*) and predicted mutation with Q5® High-Fidelity DNA Polymerase (New England Biolabs) for 25-cycles. PCR reactions cleaned up by electrophoresis on a 1% agarose gel and extracted using a Gel Extraction and PCR Cleanup Kit (IBI Scientific). Purified DNA was the submitted for targeted amplicon sequencing on an Illumina™ HiSeq® with 250 bp paired-end reads (Genewiz). All primers used for NGS library prep are reported in Supplementary Table 2.

## Target mRNA editing analysis

Paired-end reads from targeted amplicon sequencing were analyzed using Crispresso2[47]. The percentage of reads containing the predicted mutation was calculated to be the percent of reads that were trans-spliced (see Supplementary Fig. 8 for more detail).

## Western blot for Cas13 expression

HEK293 cells transfected with Cas13 expression plasmids were harvested with PBS and pelleted by centrifugation at $300 \times g$ for 5 min. Cells were lysed in RIPA buffer and treated with HALT Protease Inhibitor. Lysate was spun at $16,000 \times g$ for 5 min to remove cellular debris. Lysate was run on Mini-PROTEAN TGX gel (Bio-Rad) at 200 V for 40 min, and transferred to a membrane with the Trans-Blot Turbo Transfer System (Bio-Rad). Membranes were stained with a primary antibody against the HA to recognize the HA epitope appended to the c-terminus of each Cas protein and a secondary antibody conjugated to horse radish peroxidase (HRP). Laminin is used as a loading control (Supplementary Fig. 1).

## Flow cytometry

For flow cytometry experiments, cells were harvested 48 h post-transfection using dissociated with 100 µl .05% Trypsin (Thermo Fisher Scientific), resuspended in 500 µl 1× phosphate-buffered saline (Thermo Fisher Scientific) supplemented with 10% fetal bovine serum (Cytiva), and passed through a 100 µm filter. All flow cytometry experiments were performed on a Sony SH800, and analyzed using FlowJo. Representative gating strategy is shown in Supplementary Fig. 2.

## Transcriptomic analysis

RNA sequencing was performed on an Illumina™ HiSeq® with150bp paired-end reads (Genewiz). A previously published pipeline was used to analyze differential gene expression (https://aaronmitchd.github.io/RNA_Seq/index.html). Briefly, reads were first filtered using FastQC and TrimGalore. Abundance files were generated using Kallisto. Differential expression was calculated with DESeq2 and plotted in R. Fusion transcript detection was performed within STAR with Arriba[28].

## Gene expression by qPCR

cDNA from cells was diluted 1:10 and qPCR was performed in triplicate using LightCycler 480 SYBR Green I Mastermix (Roche). Samples containing primers specific for *LMNA*, *progerin* were normalized to *GAPDH* by ΔΔCt.

## Confocal microscopy

For cell culture experiments, HEK293 cells transfected were plated on poly-L-Lysine coated cover slips 48 h post-transfection. Cells were fixed in 200 µl 10% formalin for 30 min. They were then permeabilized using 200 µl 0.1% Trition X-100 in PBS for 20 min. 5% normal goat serum in PBS was then used as a blocking agent. Cells were treated with primary mouse anti-Lamin A antibody (1:500 Abcam, ab40567) and rabbit anti-Flag (1:500 Cell Signaling, #14793). Primary antibody was washed and cells were then treated with secondary AlexaFluor488 goat anti-rabbit (1:500 Thermo Fisher Scientific, A-11008) and AlexaFluor594 goat anti-mouse (1:500 Thermo Fisher Scientific, A-11005). Cells were mounted on slides with DAPI and imaged on a Zeiss 780 upright confocal microscope.

## Recombinant AAV vector production

Five 150mm tissue culture-treated polystyrene plates were each seeded with $15 \times 10^6$ HEK293 cells in 20 ml of media. 24 h later, cells were transfected. 60 µg pXX-680, 50 µg pXR2, and 30 µg adeno-associated viral genome plasmid were diluted in 2500 µl DMEM and mixed with 490 µl PEI diluted in 2500 µl DMEM. This mixture was vortexed and incubated at room temperature for 5 min to allow DNA/cationic lipid complexes to form and added to cells dropwise. Media containing the virus was harvested at 72 and 120 h 40% PEG8000 mixture was added to the collected media at a ratio of 1:5, incubated for 24 h and spun at $3000 \times g$ for 40 min at 4 °C. After the last media harvest, cells were also collected and spun at $300 \times g$ for 5 min. Cells were resuspended in 1 ml PBS and sonicated to lyse the cells. Lysed cell mixture was used to resuspend the PEG pellet. Virus was then purified by ultracentrifugation on an iodixanol gradient and buffer exchanged using Zebaspin desalting columns according to manufacturer's directions.

## Lentiviral vector production

100 mm tissue culture-treated polystyrene plates were seeded with $5 \times 10^6$ HEK293 cells in 10 ml of media. 24 h later, cells were transfected. 3.75 µg psPAX2, 1.5 µg pVSVG, and 5 µg lentivirus genome plasmid were diluted in 250 µl DMEM and mixed with 31.5 µl PEI diluted in 250 µl DMEM. This mixture was vortexed and incubated at room temperature for 5 min to allow DNA/cationic lipid complexes to form and added to cells dropwise. Media was changed the next morning. 48 h later, media containing the virus was collected and filtered through a 0.45 µm filter.

3 ml of PEG8000 was added to the flowthrough and incubated at 4 °C. 24 h later, PEG was precipitated by spinning this mixture at $1600 \times g$ for 1 h at 4 °C. Supernatant was removed, and the pellet was resuspended in 500 µl PBS.

## Mouse studies

Experiments involving animals were conducted with strict adherence to the guidelines for the care and use of laboratory animals of the National Institutes of Health (NIH). All experiments were approved by the Institutional Animal Care and Use Committee (IACUC) at Duke University. Mouse strains used in this study were maintained at Duke University School of Medicine with the assistance of Duke's Division of Laboratory Animal Resources (DLAR). Mice were housed in a temperature-controlled and enriched environment, with a twelve-hour light/dark cycle, and provided standard feed and water. This study utilized only male *mdx* mice on a C57BL/10 background, that were generously provided by Dr. Mai ElMallah (Duke University). It should be noted that DMD is an x-linked disorder and only male *mdx* mice were utilized in these studies accordingly.

At 8 weeks of age, $2.0 \times 10^{11}$ vector genomes ($1.0 \times 10^{11}$ vg/vector) of AAV was administered to male *mdx* mice via intramuscular injection. Mice were sacrificed at 12 weeks of age. All isolated tissues were fixed in 10% formalin overnight, and stored in sodium azide. Following fixation, heart, diaphragm, and tibialis anterior skeletal muscles were incubated in 30% sucrose overnight, embedded in O.C.T compound (Electron Microscopy Services), and frozen in liquid nitrogen-cooled isopentane. O.C.T blocks were sectioned 7 µm thick tibialis anterior on a Leica cryostat. Immunofluorescence was performed using a heat-mediated antigen retrieval method with citrate pH 6.0 buffer. Tissue sections were incubated in a blocking solution (5% normal goat serum and 0.1% Triton X-100 in 1× PBS) for 1 h at room temperature. Tissues were then incubated in rabbit anti-dystrophin (1:100; abcam 275391) and rat anti-laminin (1:400; Sigma L0663) primary antibodies diluted in blocking buffer overnight at 4 °C. The tissues were washed 3 times with 1× PBS, followed by secondary antibody incubation with goat anti-rabbit IgG AlexaFluor 647 (1:400; Invitrogen) and goat anti-rat IgG AlexaFluor594 (1:500; Invitrogen) for 1 h at room temperature. This was followed by DAPI staining (1:10,000; Thermo Fisher Scientific). Immunostained tissue sections were then mounted with ProLong Gold Antifade Mounting Media (Invitrogen).

Two stained sections from each sample (total 4 PBS control sections, 4 CRAFT injected sections, 2 wild-type positive control sections) were imaged. Whole-section multichannel images were captured with a Leica SP8 confocal microscope at 20× magnification. All exposures were chosen by trained operators based on best microscopy practices of avoiding signal saturation while maximizing the useful range of intensity values detected by the camera in the healthy positive control samples. Identical imaging settings were used for all sections.

Image quantification was performed using ImageJ v1.54f. Analysis was performed on the whole tissue area except for the border region along the tissue edges. Areas of significant artifacts (tissue folds, obvious tissue damage, mounting bubbles, etc.) were excluded from the analyzed tissue region. All images were processed using a rolling ball background correction with a 30 µm radius for both the dystrophin and laminin channels. Total fibers (anti-laminin) and dystrophin-positive fibers were manually counted in ImageJ. The dystrophin-positive fibers were normalized to the total laminin-positive fibers of each section to calculate the percent positive dystrophin fibers. Analysis was performed in GraphPad Prism v10.1.2; *p*-values were calculated with an unpaired *t*-test (ns = *p*-value > 0.05).

## Statistics and reproducibility

Where appropriate, data are represented as a mean ± standard deviation. For datasets with two groups comparisons were made between groups by students two-sided *t*-test. For datasets with at least three

groups, comparisons were made by one-way ANOVA, with Tukey's post-test. No data were excluded. These experiments were not randomized. Blinding was only performed for the analysis of confocal microscopy images. Sample sizes were chosen based on standard practices in nucleic acid editing literature.

## Reporting summary

Further information on research design is available in the Nature Portfolio Reporting Summary linked to this article.

## Data availability

High-throughput sequencing data have been deposited in the NCBI Sequence Read Archive database under the accession code PRJNA1076184. All other data associated with this study are present in the paper, supplementary materials, or sources data files. The data that support the findings of this study are available from the corresponding author upon reasonable request. Correspondence and requests for materials should be address to A.A. at Aravind.asokan@duke.edu. Source data are provided with this paper.

## Code availability

Code developed for this study is available at https://github.com/dnf97/CRAFT.

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

## Acknowledgements

We would like to acknowledge the Duke Department of Surgery, Translating Duke Health Initiative and the Duke Regenerative Medicine Center for support and funding. We also thank the Duke Cancer Institute Flow Cytometry Core and Duke Light Microscopy Core for technical assistance. We also thank the lab of Dr. Mai ElMallah for providing the *mdx* mice used in this study. Figures were created in part using Biorender and are presented here under an academic publishing license, in accordance with Biorender's terms.

## Author contributions

D.F. and A.A. conceived the study, designed experiments, and wrote the manuscript. D.F., N.A.R., and H.V.L. carried out molecular cloning, conducted experiments, and analyzed data. B.S., K.N.C, and S.M. assisted with in vitro experiments and reporter gene expression analysis. A.R.B., A.R., and S.F. conducted in vivo experiments and helped analyze data. A.M.D. contributed to high-throughput data analysis.

## Competing interests

A.A. and D.F. have filed a patent application (PCT/US2022/017959) on the subject matter of this manuscript. The authors declare no other competing interests related to the subject matter of the manuscript.
