## [Peer Review File · Nature Communications]

Reviewers' Comments:

Reviewer #1:

Remarks to the Author:

The authors have addressed most of my concerns, and they have also recognized the limitations of the study specially regarding AAV approaches, and it is appreciated that these constraints are now clear in the text. Before the acceptance of the manuscript, I would like to make some minor recommendations:

It is interesting that an increase of the RfxCas13d spacer length provides a higher trans splicing activity since that will reduce the RNA cleavage activity (Wessels et al., 2020 Nat Biot.). It would be appropriate to mention this and speculate about the differences depending on the desired approach when using RfxCas13d.

Authors stated that one of the more robust results using SMaRT achieved 40% of efficiency. Although it is clear that CRAFT outperformed SMaRT, it is also clear that authors were unable to recapitulate the level of activity of SMaRT. I think that a statement about this needs to be included in the text either during the description of the results or in the discussion.

Finally, I would recommend to add and describe Supp. Fig 10 right after Supp. Fig. 9 being part of the results section. Then in the discussion the lack of significant results using AAV can be further discussed.

Reviewer #2:

Remarks to the Author:

This revised manuscript improved in many aspects with additional data and clarifications. It is compelling to this reviewer that target/rcRNA interactions stabilized by dCas13 enhances trans-splicing and such rcRNAs require only ~30 bp of complementary regions, as compared to ~150bp for SMaRT. The authors also acknowledged that delivery of the system using AAV, especially in vivo, remains challenging and requires further optimization. The main criticism that remains is lack of a proper comparison with previous efforts using SMaRT. Using the same gRNA for CRAFT and SMaRT for reporter expression, as the authors did in this manuscript, is not sufficient, since a binding region works for CRAFT is not necessarily optimal for SMaRT. I am not asking the authors to optimize SMaRT gRNA, but it is important to compare CRAFT with the previously optimized SMaRT design in the context of endogenous genes as published in the literature. Such comparison does not have to be performed in vivo and evaluation of trans-splicing efficiency would be sufficient (does not have to be correction disease phenotypes), and quite a few studies have been published (including DMPK as tested in this paper). As the authors stated, in the current form, it might be easier to design a working gRNA using CRAFT, but it is unclear whether CRAFT can really achieve superior trans-splicing efficiency than SMaRT, if the gRNA design were optimized for the latter. This is an important point for the paper, especially given the delivery challenge of CRAFT as compared to SMaRT.

Reviewer #3:

Remarks to the Author:

The revised manuscript has largely improved. Unfortunately, the in vivo PoC with AAV driven CRAFT failed, but this was somewhat expected. Accordingly, the authors toned down the claims. Nevertheless, the manuscript is an important contribution to the field and certainly merits

publication in Nature Comms, from my site also without further alteration.

Response To Comments

Thank you again for submitting your revised manuscript "Repurposing CRISPR-Cas13 systems for robust mRNA trans-splicing" to Nature Communications. We have now received reports from the reviewers who evaluated the original version. On the basis of their comments (copied below), we have decided to invite an additional revision of your work.

You will see that, while the reviewers find that your revisions improved the manuscript, some important points remain to be addressed regarding Reviewer #2's comments regarding an evaluation of trans-splicing efficiency. Please revise your manuscript, addressing all the remaining issues raised by the reviewers.

We would like to thank you and the reviewers for the opportunity to address remaining issues and submit our revised manuscript.

REVIEWER COMMENTS

Reviewer #1 (Remarks to the Author):

The authors have addressed most of my concerns, and they have also recognized the limitations of the study specially regarding AAV approaches, and it is appreciated that these constrains are now clear in the text.

We thank the reviewer for the comments and earlier recommendations that helped improve the manuscript.

Before the acceptance of the manuscript, I would like to make some minor recommendations:

It is interesting that an increase of the RfxCas13d spacer length provides a higher trans splicing activity since that will reduce the RNA cleavage activity (Wessels et al., 2020 Nat Biot.). It would be appropriate to mention this and speculate about the differences depending on the desired approach when using RfxCas13d.

We have now mentioned this as recommended:

"It is plausible that different applications for Cas13 may benefit from extended Watson-Crick base pairing as that may improve the affinity for a given target. However, this may come at the detriment of the cleavage activity for applications of catalytically active cas13 as target recognition and cleavage are dynamically linked."

Authors stated that one of the more robust results using SMaRT achieved 40% of efficiency. Although is clear that CRAFT outperformed SMaRT, it is also clear that authors were unable to recapitulate the level of activity of SMaRT. I think that a statement about this needs to be included in the text either during the description of the results or in the discussion.

We have now included a statement outlining the differences between the published results and ours.

Finally, I would recommend to add and describe Supp. Fig 10 right after Supp. Fig. 9 being part of the results section. Then in the discussion the lack of significant results using AAV can be further discussed.

This change has now been implemented as recommended.

Reviewer #2 (Remarks to the Author):

This revised manuscript improved in many aspects with additional data and clarifications. It is compelling to this reviewer that target/rcRNA interactions stabilized by dCas13 enhances trans-splicing and such rcRNAs require only ~30 bp of complementary regions, as compared to ~150bp for SMaRT. The authors also acknowledged that delivery of the system using AAV, especially in vivo, remains challenging and requires further optimization.

We thank the reviewer for these comments.

The main criticism that remains is lack of a proper comparison with previous efforts using SMaRT. Using the same gRNA for CRAFT and SMaRT for reporter expression, as the authors did in this manuscript, is not sufficient, since a binding region works for CRAFT is not necessary optimal for SMaRT. I am not asking the authors to optimize SMaRT gRNA, but it is important to compare CRAFT with the previously optimized SMaRT design in the context of endogenous genes as published in the literature. Such comparison does not have to be performed in vivo and evaluation of trans-splicing efficiency would be sufficient (does not have to be correction disease phenotypes), and quite a few studies have been published (including DMPK as tested in this paper). As the authors stated, in the current form, it might be easier to design a working gRNA using CRAFT, but it is unclear whether CRAFT can really achieve superior trans-splicing efficiency than SMaRT, if the gRNA design were optimized for the latter. This is an important point for the paper, especially given the delivery challenge of CRAFT as compared to SMaRT.

We have now incorporated a direct comparison of SMaRT vs. CRAFT in the context of the endogenous human dmd transcript in the main text/fig. 5. These results compare 3 different unoptimized guides for either approach and corroborate that dead Cas13 can indeed augment trans-splicing efficiency of endogenous transcripts. With regard to the dmpk target, we note that targeting the exact same intron (as has been tested and published [ref 32]) with CRAFT enables greater trans-splicing efficiency than without dCas13 (figure 6e). Unfortunately, we were unable to evaluate the endogenous rho target, which would require a retinal cell line. When taken together with the endogenous dmd and dmpk examples, as well as multiple intronic targets in the context of GFP we feel that substantial evidence is in place to corroborate that CRAFT can improve trans-splicing efficiency. We do agree, as the reviewer points out, that optimizing SMaRT design could continue to improve that approach. However, our intent in this

manuscript was to explore whether we could repurpose type-VI CRISPR systems and we are confident that the results unequivocally support such.

Reviewer #3 (Remarks to the Author):

The revised manuscript has largely improved. Unfortunately, the in vivo PoC with AAV driven CRAFT failed, but this was somewhat expected. Accordingly, the authors toned down the claims. Nevertheless, the manuscript is an important contribution to the field and certainly merits publication in Nature Comms, from my site also without further alteration.

We appreciate the prior input which helped improve the manuscript.